# Improving Performance of Al_2_O_3_/AlN/GaN MIS HEMTs via In Situ N_2_ Plasma Annealing

**DOI:** 10.3390/mi14061100

**Published:** 2023-05-23

**Authors:** Mengyuan Sun, Luyu Wang, Penghao Zhang, Kun Chen

**Affiliations:** 1State Key Laboratory of ASIC and System, School of Microelectronics, Fudan University, Shanghai 200433, China; 20212020003@fudan.edu.cn (M.S.); wangly20@fudan.edu.cn (L.W.); phzhang19@fudan.edu.cn (P.Z.); 2Shanghai Integrated Circuit Manufacturing Innovation Center Co., Ltd., Shanghai 200433, China

**Keywords:** fully recessed-gate MIS HEMTs, AlN film, plasma annealing, threshold voltage

## Abstract

A novel monocrystalline AlN interfacial layer formation method is proposed to improve the device performance of the fully recessed-gate Al_2_O_3_/AlN/GaN Metal-Insulator-Semiconductor High Electron Mobility Transistors (MIS-HEMTs), which is achieved by plasma-enhanced atomic layer deposition (PEALD) and in situ N_2_ plasma annealing (NPA). Compared with the traditional RTA method, the NPA process not only avoids the device damage caused by high temperatures but also obtains a high-quality AlN monocrystalline film that avoids natural oxidation by in situ growth. As a contrast with the conventional PELAD amorphous AlN, *C-V* results indicated a significantly lower interface density of states (*D*_it_) in a MIS *C-V* characterization, which could be attributed to the polarization effect induced by the AlN crystal from the X-ray Diffraction (XRD) and Transmission Electron Microscope (TEM) characterizations. The proposed method could reduce the subthreshold swing, and the Al_2_O_3_/AlN/GaN MIS-HEMTs were significantly enhanced with ~38% lower on-resistance at *V*_g_ = 10 V. What is more, in situ NPA provides a more stable threshold voltage (*V*_th_) after a long gate stress time, and Δ*V*_th_ is inhibited by about 40 mV under *V*_g,stress_ = 10 V for 1000 s, showing great potential for improving Al_2_O_3_/AlN/GaN MIS-HEMT gate reliability.

## 1. Introduction

Gallium nitride (GaN) and its related wide-band gap compound semiconductors have been considered candidates for the next generation of RF and power conversion applications [1,2,3,4,5]. Compared with Si, high electron mobility transistors (HEMTs) based on AlGaN/GaN heterostructures have excellent performance due to their inherent high breakdown strength, low on-resistance, and high temperature operating capability [6,7]. In addition to these inherent advantages, there is also great interest in the possibility of growing GaN-based semiconductors on large-area (up to 200 mm) and low-cost Si substrates due to the large market potential and the possibility of integrating GaN power switches with Si CMOS technology [2]. The conventional AlGaN/GaN HEMTs operate in depletion mode (D-mode) because of the high-density 2DEG induced by the polarization effect. However, in order to reduce power loss during switching and to simplify circuit configuration, normally-off devices are necessary for power applications. Many technologies that could enable devices to achieve enhancement mode (E-mode) have been proposed, including fluorine ion treatment [8,9], p-GaN gate [10], thin AlGaN barrier [11], recessed gate [12,13], and so on. While the F ion implantation method appeared earlier, the F ion is easy to diffuse in the barrier layer at high temperatures, resulting in an unstable device threshold voltage. The thin AlGaN barrier structure will reduce the polarization effect of the draft region, resulting in a decrease in two-dimensional electron gas (2DEG) concentration and device output characteristics. p-GaN gate structure requires very precise etching conditions [14] and etch-induced damage is inevitable to the draft region. Among them, recessed gate HEMTs are considered one of the most promising approaches.

In order to make the device normally off, the polarization-induced charges will be eliminated when the AlGaN barrier under the gate region is fully removed. The barrier-removing process can introduce extra etching damage and surface states. Therefore, gate dielectric is significant for the recessed-gate structure because it can decrease off-state gate leakage and driver losses. At the same time, with the merits of suppressed gate leakage and enlarged gate swing, AlGaN/GaN metal-insulator-semiconductor high electron mobility transistors (MIS-HEMTs) are highly preferred over the conventional Schottky-gate HEMTs for high-voltage power switches. However, the insertion of gate dielectric creates an additional dielectric/AlGaN/GaN interface, where a high density of interface traps usually exists [15]. The Al_2_O_3_ film grown by atomic layer deposition (ALD) is commonly used as the gate dielectric in the fabrication of AlGaN/GaN MIS-HEMTs [16]. However, under high fabrication process temperatures and high operation voltage stress conditions, the oxygen element in ALD Al_2_O_3_ may diffuse to the surface of the AlGaN barrier layer, causing reliability concerns [17].

In terms of interface quality, Hinkle et al. [18] found that natural oxides (Ga_2_O_3_) on the surface of (Al)GaN compound semiconductors are the main reason for Fermi level pinning and high interface trap density (*D*_it_) in GaN-based transistors. Robertson [19] proposed that natural defects such as Ga-suspended bonds at the oxide/(Al)GaN interface also restrict Fermi level variation. In addition to traditional natural oxide removal methods, nitridation interfacial layer (NIL) ahead of the gate dielectric deposition has been adopted in AlGaN/GaN HEMTs, such as AlN deposition [20], thermal nitridation [21], and remote plasma nitridation [22,23,24]. Implemented in a plasma-enhanced atomic layer deposition (PEALD) system, NIL is an excellent choice for interface quality improvement. It is because PEALD not only can achieve precise deposition thickness and avoid surface damage caused by plasma but also facilitates implementing the function of in situ dielectric deposition.

Monocrystalline AlN film has been reported as an interfacial dielectric layer in MIS-HEMTs, which can improve interface quality and device reliability [25]. Generally, high-temperature (over 600 °C) processes such as molecular beam epitaxy (MBE) and metal-organic vapor deposition (MOCVD) are considered necessary for the formation of high-quality AlN crystal [26]. Nevertheless, such a high temperature is not compatible with the subsequent fabrication process [27,28]. Although the ALD technique could facilitate the growth of AlN at a lower temperature (about 300 °C), the amorphous film still needs high temperature rapid thermal annealing (RTA) to transform to the monocrystalline state.

In this work, a novel in situ AlN crystal interfacial layer formation process is proposed to obtain a high-quality Al_2_O_3_/AlN/GaN interface. Compared with the traditional RTA method, the NPA process not only avoids the device damage caused by high temperatures but also obtains a high-quality AlN monocrystalline film that avoids natural oxidation by in situ growth. Namely, the post PEALD in situ N_2_ plasma annealing (NPA) process not only promotes the crystallization of the AlN interfacial layer but also significantly suppresses interface states and reduces *V*_th_ shift after long-term gate stress as compared to fully recessed-gate MIS-HEMTs fabricated with conventional amorphous AlN.

## 2. Device Structure and Fabrication

For the fabrication of fully recessed-gate MIS-HEMTs, the epitaxial structure was grown on a 6-in Si (111) wafer by MOCVD. The epitaxial III-Nitride layers were composed of a 5 μm C-doped buffer layer, a 180 nm GaN channel, and a 20 nm Al_0.22_Ga_0.78_N barrier layer. The density and mobility of 2DEG were 8.5 × 10^12^ cm^−2^ and 1960 cm^2^/Vs, respectively, by Hall measurement at room temperature.

Figure 1a shows the schematic of the fully recessed-gate MIS-HEMTs, in which gate lengths of 4 μm and gate widths of 50 μm are prepared for the following characterization. After device isolation by Cl-based inductively coupled plasma (ICP) deep etch, a 50-nm SiN_x_ passivation stack was deposited by ICP-CVD. After selectively removing the passivation layers in the gate window by F-based ICP dry etching, the gate recess process was performed by O_2_ and BCl_3_ atomic layer etch (ALE) technology at an etch rate of 0.75 nm/cyc [29,30]. As shown in Figure 1b, a total recess depth of ∼20 nm was reached after 26 cycles of ALE, indicating complete removal of the barrier layer. Then, the AlN insertion layer was deposited by PEALD. Trimethylaluminum and NH_3_ were used as the metal precursor and the N source, respectively. The purge gas was high-purity N_2_. The plasma RF power and chamber temperature were set at 60 W and 250 °C, respectively. In order to avoid excessive polarization charge at the AlN/GaN interface, the thickness of the AlN film with 20 cycles of ALD was nominally 1.5 nm. An in situ NPA process was first applied to make the amorphous AlN transform to the crystalline form, as shown in Table 1. Considering that thin AlN was the interface layer, an RF of 200 W and a process time of 300 s were chosen for high quality and low damage requirements. Afterwards, a 20 nm Al_2_O_3_ layer was in situ deposited by ALD, followed by 500 °C RTA for 90 s to eliminate dangling bonds. Finally, the Ti/Al/Ni/Au ohmic contact and Ni/Au gate were fabricated, respectively. In addition, MOS diodes for capacitance–voltage (*C-V*) tests were also prepared on the same wafer, as shown in Figure 1c. For comparison, MIS HEMTs with two different properties of the AlN interfacial layer were fabricated, which are also distinguished as Scheme I and Scheme II.

Grazing incidence X-ray diffraction (GIXRD) was used to investigate the crystallization characteristics of the AlN, as shown in Figure 1d. It can be observed that the AlN peak appeared in the (0002) orientation. The diffraction peak intensity after NPA was significantly enhanced in Scheme II. This indicates that the proposed process can promote the crystallization of AlN films on the GaN substrate.

## 3. Results and Discussion

The multi-frequency capacitance–voltage characteristics of the MOS diodes are plotted in Figure 2a,b. With the frequency varying from 10 kHz to 10 MHz, the MOS diodes with an N_2_ plasma-enhanced AlN crystal interfacial layer show much smaller frequency dispersion compared to Scheme II, indicating an improved interface with a lower trap density. The insets in Figure 2a,b show the *C-V* hysteresis characteristics at the frequency of 1 MHz. The shift of flat-band voltage is 53 mV and 129 mV for the MOS diodes in Schemes I and II, respectively. Trap density (*D*_it_) can be obtained from multi-frequency *C-V* curve frequency dispersion [31]. For Scheme I, the measured Δ*V*_FB_ between 100 kHz and 1 MHz was 40 mV, indicating 9.7 × 10^11^ cm^−2^ eV^−1^ of trap states with a time constant in the range of 0.16~16 μs. The corresponding trap densities for Scheme II were 8 × 10^12^ cm^−2^ eV^−1^. It can be seen that the MOS diodes in Scheme I have a lower *V*_FB_. This is because the AlN crystal interfacial layer had an enhanced polarization effect, and more polarization charges were generated at the interface of AlN/GaN [17]. The cross-sectional TEM micrographs of Al_2_O_3_/AlN/GaN interfaces in the recessed region are shown in Figure 2c,d. A sharp monocrystal interfacial layer is formed through the NPA process. In contrast, Scheme II exhibits a rough interface.

Figure 3a shows the fully recessed-gate MIS-HEMTs transfer characteristics at *V*_d_ = 10 V of Scheme I and Scheme II, respectively. A normally-off operation with a *V*_th_ of 1.6 V is achieved. The subthreshold swing of Scheme I is much lower than Scheme II. What is more, the device of Scheme I exhibited well-suppressed off-state gate leakage compared with Scheme II at *V*_d_ = 10 V. The max saturated drain current (*I*_sat_) is 371 mA/mm and 301 mA/mm at *V*_g_ = 8 V of Scheme I and Scheme II, respectively, as illustrated in Figure 3b. The extracted on-resistance (*R*_ON_) of Scheme I and Scheme II are 10.1 Ω·mm and 15.93 Ω·mm at *V*_g_ = 10 V, respectively.

The threshold voltage (*V*_th_) instability after a positive forward-reverse gate sweep or *V*_th_ shift during a positive gate bias stress, which is generally referred to as positive bias temperature instability (PBTI), has been reported for different gate dielectrics [32,33,34]. The PBTI represents serious reliability issues in fully recessed-gate MIS HEMTs for E-mode applications since a high-gate overdrive (*V*_g_−*V*_th_) is needed for fast switching [35]. Figure 4a shows the curves of threshold voltage shift (Δ*V*_th_) versus stress period (*t*_stress_) at *V*_g,stress_ = 6 V, 8 V, and 10 V, respectively. We observe that Δ*V*_th_ increases with *V*_g,stress_, and *t*_stress_ increasing. After being stressed for 1000 s, Δ*V*_th_ exhibits 90 mV and 130 mV for *V*_g,stress_ = 10 V of Scheme I and Scheme II, respectively, which results from the high quality AlN crystal interfacial layer reducing the defect density and maintaining a more stable threshold voltage (*V*_th_). After the stress phase, all devices are immediately biased at *V*_g,recovery_ = 0 V to record the *V*_th_ recovery at room temperature (Figure 4b). It is difficult to recover completely even when biased at *V*_g,recovery_ = 0 V, indicating that a higher density of trap states may be introduced by high overdrive voltage.

## 4. Conclusions

In summary, a novel post-AlN growth in situ NPA process is proposed to improve device performance. The newly proposed in situ NPA process could effectively promote the crystallization of the AlN interfacial layer. Compared with the traditional RTA method, the NPA process not only avoids high temperature-induced damage to the devices but also produces a high-quality AlN monocrystalline film that avoids natural oxidation by in situ growth. As a contrast with the conventional PELAD amorphous AlN, *C-V* results indicated a significantly lower interface density of states (*D*_it_) in a MIS diode *C-V* characterization, which could be attributed to the polarization effect induced by the AlN crystal. The NPA process was helpful for the subthreshold swing reduction, and the Al_2_O_3_/AlN/GaN MIS-HEMTs were significantly enhanced with ~38% lower on-resistance at *V*_g_ = 10 V. What is more, in situ NPA provides a more stable *V*_th_ after a long gate stress time, and Δ*V*_th_ is inhibited by about 40 mV under *V*_g,stress_ = 10 V for 1000 s, showing great potential for improving Al_2_O_3_/AlN/GaN MIS-HEMT gate reliability.

## Figures and Tables

**Figure 1 micromachines-14-01100-f001:**
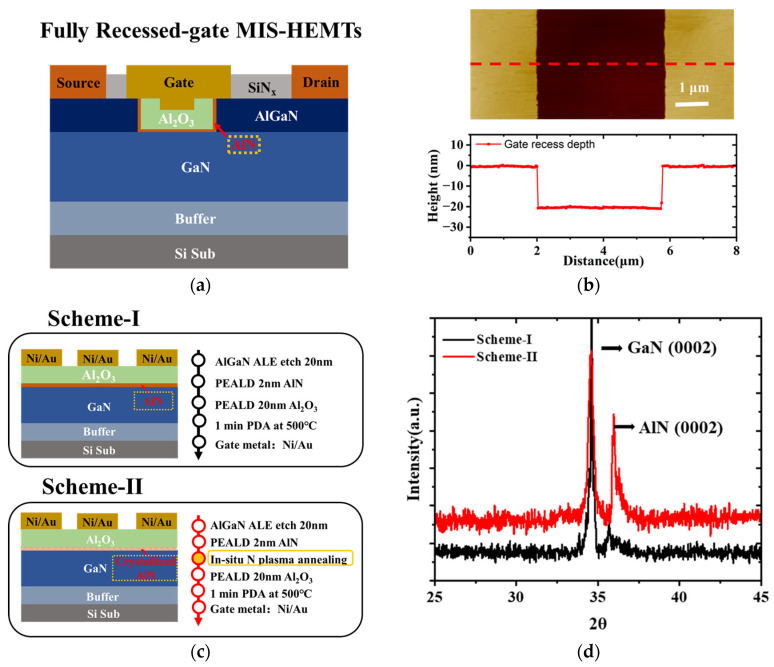
(**a**) Schematic cross-sectional view of normally-off Al_2_O_3_/AlN/GaN MIS-HEMT on silicon substrate; (**b**) AFM measurement of the trench profile along the recessed window; (**c**) The MOS diodes for capacitance–voltage (*C-V*) tests; (**d**) XRD spectrum of AlN (0002) and GaN (0002).

**Figure 2 micromachines-14-01100-f002:**
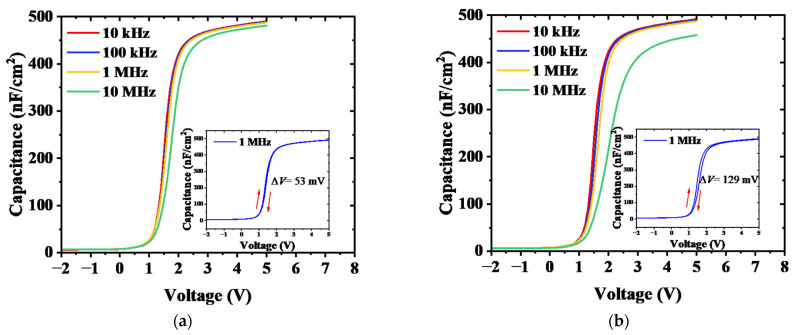
Multi-frequency *C-V* curves for diodes in Scheme I (**a**) and Scheme II (**b**). The insets were the hysteresis curves at 1 MHz and cross-sectional TEM images for the MOS structure in Scheme I (**c**) and Scheme II (**d**).

**Figure 3 micromachines-14-01100-f003:**
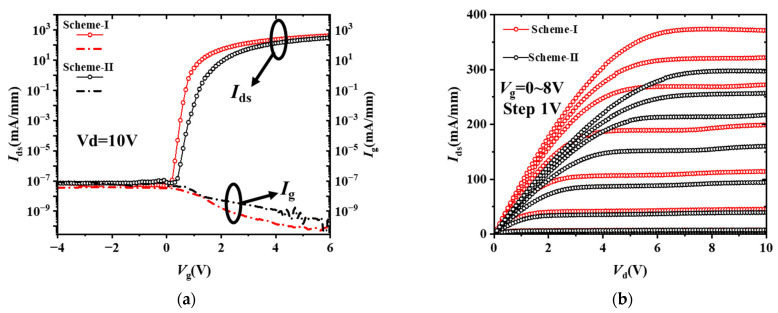
Device performance of the fabricated fully recessed-gate MIS HEMTs (**a**) transfer characteristics in semi-logarithm scale (insert graph linear scale) and (**b**) output characteristics.

**Figure 4 micromachines-14-01100-f004:**
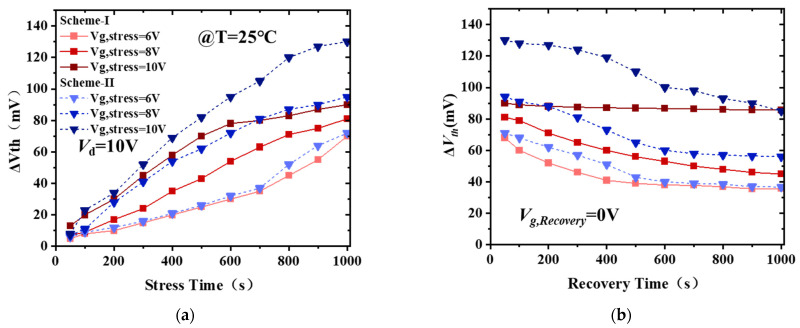
Changes of (**a**) Δ*V*_th_ versus *t*_stress_ under different gate voltage stresses at room temperature and (**b**) Δ*V*_th_ recovery at *V*_g,recovery_ = 0 V.

**Table 1 micromachines-14-01100-t001:** Conditions of N_2_ plasma annealing for AlN properties.

Power/W	Time/s	AlN Property
100	100	amorphous
100	300	amorphous
100	500	amorphous
200	100	Weak signal in AlN (0002)
**200**	**300**	**monocrystalline**
200	500	monocrystalline
300	100	monocrystalline
300	300	monocrystalline
300	500	Weak signal in AlN (0002)

## Data Availability

Not applicable.

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
