# Peer review of "Improving Performance of Al2O3/AlN/GaN MIS HEMTs via In Situ N2 Plasma Annealing"

_micromachines, 2023, doi:10.3390/mi14061100_

Round 1

Reviewer 1 Report

This manuscript demonstrates the improved performance of Al2O3/AlN/GaN MIS HEMTs via in-situ N2 plasma annealing. The device performance is characterized by C-V, I-V measurements, and threshold voltage stability under stress. The manuscript is well written, and the topic would be of interest to the readers. I suggest the publication in micromachines after addressing the following comments.

1.       The authors should cite previous studies on N2 annealing in Al2O3/AlN/GaN MIS HEMT and compare the results to the previous literature. For example, Chen et al. Phys. Status Solidi A212, No. 5, 1059–1065 (2015), He et al., Phys. Status Solidi A2019,216, 1900115, Romero et al., IEEE ELECTRON DEVICE LETTERS, VOL. 29, NO. 3, MARCH 2008.

2.       The novelty of the present study should be highlighted.

3.       How do N2 annealing process parameters (temperature/time/plasma power) affect the device performance?

4.       Figure 1 quality needs to be improved. The text in Figure 1c is hard to read.

5.       Figure 4b is missing Y-axis.

6.       It would be helpful to summarize the major performance improvement by N2 plasma treatment with numbers (for example,  improvement of xxx value) in the conclusion/abstract.

Please double-check the use of subscripts (VFB line 96, line 99);

Abbreviations in the abstract should be avoided.

Author Response

We really appreciate your favorite consideration and insightful comments concerning our manuscript entitled “Improving performance of Al2O3/AlN/GaN MIS HEMTs via in-situ N2 plasma annealing”. Those comments are very valuable and helpful for improving the quality and readability of our paper. We have studied the comments carefully and revised the paper accordingly as below.

Point 1: The authors should cite previous studies on N2 annealing in Al2O3/AlN/GaN MIS HEMT and compare the results to the previous literature. For example, Chen et al. Phys. Status Solidi A212, No. 5, 1059–1065 (2015), He et al., Phys. Status Solidi A2019,216, 1900115, Romero et al., IEEE ELECTRON DEVICE LETTERS, VOL. 29, NO. 3, MARCH 2008.

Response 1: Thank you very much for your good suggestion. Three suggested literatures have been cited in the article to enrich the content.

Corresponding change in manuscript: Yes.

Location of Change: Introduction paragraph 1 & 2 and reference[15][13][23].

Point 2: The novelty of the present study should be highlighted.

Response 2: Thank you very much for your comment. It has been emphasized in abstract, introduction and conclusion that compared with the traditional RTA method, NPA process not only avoids the device damage caused by high temperature, but also obtains high quality AlN monocrystalline film which avoids natural oxidation by in-situ growth method.

Corresponding change in manuscript: Yes

Location of Change: Abstract, Line87~90, conclusion.

Point 3: How do N2 annealing process parameters (temperature/time/plasma power) affect the device performance?

Response 3: Thank you very much for your valuable suggestion. The effect of N2 annealing condition on the properties of AlN is added in this paper, and the optimal scheme under RF of 200 W and process time of 300 s is selected and compared with the amorphous AlN interfacial layer device.

Corresponding change in manuscript: Yes (A table has been added for comparison )

Location of Change: Line130~131.

Point 4: Figure 1 quality needs to be improved. The text in Figure 1c is hard to read.

Response 4: Thank you very much for your suggestion. Higher quality images and bigger text of Figure 1c were updated in the manuscript.

Corresponding change in manuscript: Yes

Location of Change: Figure1 (page 3)

Point 5: Figure 4b is missing Y-axis.

Response 5: Thank you very much for your suggestion. The Y-axis annotations are supplemented.

Corresponding change in manuscript: Yes

Location of Change: Figure4 (page 6)

Point 6: It would be helpful to summarize the major performance improvement by N2 plasma treatment with numbers (for example, improvement of xxx value) in the conclusion/abstract.

Response 6: Thank you very much for your suggestion. A quantitative comparison of the interface states (Dit) and device performance under different processes has been supplemented in the abstract and conclusion.

Corresponding change in manuscript: Yes

Location of Change: Abstract & Conclusion.

Reviewer 2 Report

The manuscript represented a post growth in-situ annealing technique using N2 plasma to improve the HEMT device performance. They successfully demonstrated the stable threshold voltage with reduced on-resistance of the developed MIS-HEMT device. Their study is accompanied with sufficient experimental data.

There are minor grammatical corrections required in the manuscript.

Author Response

We really appreciate your favorite consideration and insightful comments concerning our manuscript entitled “Improving performance of Al2O3/AlN/GaN MIS HEMTs via in-situ N2 plasma annealing”. Those comments are very valuable and helpful for improving the quality and readability of our paper. We have studied the comments carefully and revised the paper accordingly as below.

Point 1: There are minor grammatical corrections required in the manuscript.

Response 1: Thank you very much for your comment. The grammar of the paper has been rechecked according to your requirements, and the abstract, introduction and conclusion have been rewritten.

Reviewer 3 Report

1. In the Introduction.........."Gallium Nitride is widely adopted in power conversion, radio frequencies, and pho- 24 toelectric applications nowadays. The conventional AlGaN/GaN High electron mobility 25 transistors (HEMTs) are operating in depletion mode (D-Mode) because of the high-den- 26 sity 2DEG induced by the polarization effect".............Citation required

The authors can refer the below manuscripts

a) https://ieeexplore.ieee.org/document/8962057

b) https://www.sciencedirect.com/science/article/pii/S1369800122005145

2. The dimensions of the gate (length and width ) are missing in the manuscript.

3. The authors used Vds and Vgs in the text and Vd and Vg in figures. It is confusing..maintain uniformity in representing variables.

4. Conclusion is not sufficient. Can be added the applications of GaN HEMTs.

5. Introduction section can be updated by adding the challenges in the fabrication of GaN HEMTS. Number of references used in the article is very less. Refer the below articles for describing the recent trends in developing GaN BASED devices. (Refer https://ieeexplore.ieee.org/document/10038498, https://ieeexplore.ieee.org/document/10075406, https://www.sciencedirect.com/science/article/pii/S2773012322001303, 

https://ieeexplore.ieee.org/document/10083134

Author Response

We really appreciate your favorite consideration and insightful comments concerning our manuscript entitled “Improving performance of Al2O3/AlN/GaN MIS HEMTs via in-situ N2 plasma annealing”. Those comments are very valuable and helpful for improving the quality and readability of our paper. We have studied the comments carefully and revised the paper accordingly as below.

Point 1: In the Introduction.........."Gallium Nitride is widely adopted in power conversion, radio frequencies, and photoelectric applications nowadays. The conventional AlGaN/GaN High electron mobility transistors (HEMTs) are operating in depletion mode (D-Mode) because of the high-density 2DEG induced by the polarization effect".............Citation required

The authors can refer the below manuscripts

  1. a) https://ieeexplore.ieee.org/document/8962057
  2. b) https://www.sciencedirect.com/science/article/pii/S1369800122005145

 Response 1: Thank you very much for your comment. The abstract has been reorganized according to your request, and the recommended literature has been cited.

Corresponding change in manuscript: Yes

Location of Change: Introduction and reference[2][3].

Point 2: The dimensions of the gate (length and width ) are missing in the manuscript.

Response 2: Thank you very much for your insightful comment. “The gate lengths featuring 4 μm and the gate width is 50 μm” is highlight in the manuscript.

Corresponding change in manuscript: Yes

Location of Change: Line 103~104.

Point 3: The authors used Vds and Vgs in the text and Vd and Vg in figures. It is confusing. maintain uniformity in representing variables.

 Response 3: Thank you very much for your comment. The notes in the article and the picture have been unified, both are Vd and Vg。

Corresponding change in manuscript: Yes

Location of Change: Line 152~159.

Point 4: Conclusion is not sufficient. Can be added the applications of GaN HEMTs.

Response 4: Thank you very much for your good suggestion. The conclusions have been rewritten and quantified the improvement of NPA process.

Corresponding change in manuscript: Yes

Location of Change: Conclusion.

Point 5: Introduction section can be updated by adding the challenges in the fabrication of GaN HEMTS. Number of references used in the article is very less. Refer the below articles for describing the recent trends in developing GaN BASED devices.

(Refer https://ieeexplore.ieee.org/document/10038498, https://ieeexplore.ieee.org/document/10075406, https://www.sciencedirect.com/science/article/pii/S2773012322001303,

https://ieeexplore.ieee.org/document/10083134

Response 5: Thank you very much for your good suggestion. The introduction has been reorganized according to your request, and the recommended literature has been cited.

Corresponding change in manuscript: Yes

Location of Change: Introduction and reference[4][34][5][9].
